# The reaction of hydroxyl and methylperoxy radicals is not a major source of atmospheric methanol

Rebecca L. Caravan [1], M. Anwar H. Khan [2], Judit Zádor[1], Leonid Sheps[1], Ivan O. Antonov[1], Brandon Rotavera [1], Krupa Ramasesha[1], Kendrew Au[1], Ming-Wei Chen[1], Daniel Rösch[1], David L. Osborn [1], Christa Fittschen [3], Coralie Schoemaecker[3], Marius Duncianu[4], Asma Grira[4], Sebastien Dusanter[4], Alexandre Tomas[4], Carl J. Percival[5], Dudley E. Shallcross[2] & Craig A. Taatjes [1]

Methanol is a benchmark for understanding tropospheric oxidation, but is underpredicted by up to 100% in atmospheric models. Recent work has suggested this discrepancy can be reconciled by the rapid reaction of hydroxyl and methylperoxy radicals with a methanol branching fraction of 30%. However, for fractions below 15%, methanol underprediction is exacerbated. Theoretical investigations of this reaction are challenging because of intersystem crossing between singlet and triplet surfaces – ~45% of reaction products are obtained via intersystem crossing of a pre-product complex – which demands experimental determinations of product branching. Here we report direct measurements of methanol from this reaction. A branching fraction below 15% is established, consequently highlighting a large gap in the understanding of global methanol sources. These results support the recent high-level theoretical work and substantially reduce its uncertainties.

[1] Combustion Research Facility, Mailstop 9055, Sandia National Laboratories, Livermore, CA 94551, USA. [2] School of Chemistry, Cantock's Close, University of Bristol, Bristol BS8 1TS, UK. [3] Université Lille, CNRS, UMR 8522–PC2A–Physicochimie des Processus de Combustion et de l'Atmosphère, 59000 Lille, France. [4] IMT Lille Douai, Université Lille, Département Sciences de l'Atmosphère et Génie de l'Environnement (SAGE), 59000 Lille, France. [5] Jet Propulsion Laboratory, California Institute of Technology, 4800 Oak Grove Drive, Pasadena, CA 91109, USA. Correspondence and requests for materials should be addressed to R.L.C. (email: rcarava@sandia.gov) or to D.E.S. (email: d.e.shallcross@bris.ac.uk) or to C.A.T. (email: cataatj@sandia.gov)

The hydroxyl radical, OH, sometimes called the tropospheric detergent, is an essential oxidant[1] in Earth's lower atmosphere[2]. In the absence of substantial anthropogenic contributions, dominant atmospheric sinks of OH are reactions with CO and CH$_4$. The reaction of OH with CH$_4$ (1) yields the simplest and most abundant atmospheric alkylperoxy radical, methylperoxy (CH$_3$OO)[3]

$$OH + CH_4 (+O_2) \rightarrow CH_3OO + H_2O \qquad (1)$$

Steady-state concentrations of methylperoxy range between 1–20 ppt;[3] atmospheric sinks include reaction with NO, HO$_2$, and self- and cross-reactions with other peroxy radical species[4]. The latter reactions lead to methanol production of 48 teragrams (Tg) per year[5] and are consequently an important source of atmospheric methanol, in particular over remote regions where primary emission sources, such as plant growth, plant decay, and anthropogenic sources are negligible[6,7].

Methanol concentrations range from 1–15 ppbv in the continental boundary layer and 0.1–1 ppbv in the remote troposphere[5,8,9]. Oxidation of methanol forms species including CO, formaldehyde, and tropospheric ozone[10], and reactions of alcohols may have subtle, indirect effects in the formation of secondary organic aerosols[11], therefore impacting the tropospheric oxidising capacity, air quality and human health. Atmospheric methanol abundances are dominated by direct emissions but sources also include oxidation pathways of methane and other volatile organic species. Methanol is thus a benchmark for the performance of atmospheric models. Despite inclusion of multiple methanol production pathways, global atmospheric chemical models are presently unable to reconcile the modelled and measured methanol abundances over remote regions[5], and so other production pathways have been sought.

Until recently, the coupling between OH and CH$_3$OO had not been investigated, despite the large rate coefficient recommended by Tsang and Hampson[12] ($k = 1 \times 10^{-10}$ cm$^3$ s$^{-1}$) and the suggestion that a major product would be methanol (CH$_3$OH)[13]. Four likely product channels exist:

$$CH_3OO + OH \rightarrow CH_3O + HO_2 \qquad (2)$$

$$CH_3OO + OH \rightarrow CH_2OO + H_2O \qquad (3)$$

$$CH_3OO + OH \rightarrow CH_3OH + O_2 \qquad (4)$$

$$CH_3OO + OH (+M) \rightarrow CH_3OOOH (+M) \qquad (5)$$

Recent experiments[14,15] established a high rate coefficient for reactions of OH with methylperoxy (CH$_3$OO), between $1$–$2 \times 10^{-10}$ cm$^3$ s$^{-1}$, with similar rate coefficients for larger alkylperoxy radicals[16,17]. The branching fractions for the product channels of the OH + CH$_3$OO reaction have been estimated through theoretical approaches by Müller et al.[18] and channels (2) and (3) probed experimentally by Yan et al.[14] and Assaf et al.[19]. To date, no experimental studies have directly measured $\phi_{CH3OH}$.

Substantial mechanistic insight into this reaction is given by the high-level ab initio calculations in Müller et al.[18], which characterise key stationary points on the reaction potential energy surface including the three bimolecular product channels (2–4), the trioxide association product (CH$_3$OOOH) (5) and a pre-product complex. Müller et al. ascertained product branching ratios through RRKM calculations, which, notably, identify triplet entrance routes as "entirely negligible," and show the dominance of the singlet trioxide intermediate, which can rapidly convert to

the pre-product complex (CH$_3$O…HOO), which has only a 40 ps lifetime at its initial energies. The coupling via intersystem crossing (ISC) of the singlet and triplet states of the product complex affects the product branching. The singlet state primarily undergoes rapid H-bond scission to yield bimolecular products CH$_3$O + HO$_2$ (2) with a small (~5%) component dissociating to CH$_3$OH and O$_2$ ($^1\Delta$). The triplet state has competing pathways: direct and indirect CH$_3$O + HO$_2$ production (the latter via ISC back to the singlet state) or rearrangement and subsequent decomposition to CH$_3$OH + O$_2$ ($^3\Sigma_g^-$). The multiple favourable routes to (2) serve to facilitate high yields of HO$_2$ and CH$_3$O; branching from the triplet state of the pre-product complex to methanol (4) is calculated to be about twice as favourable (~10%) as its formation from the singlet surface. However, Müller et al[18]. estimated that the uncertainty in the stationary point energies and in the ISC probability gave uncertainties of a factor of 3.5 in the branching fractions. Dramatically different tropospheric effects are encompassed by the upper and lower limits of the methanol yield, $\phi_{CH3OH}$, given by Assaf et al.[19] (0–40%), Müller et al.[18] (2–30%) and Ferracci et al.[20] (0–40%).

The potential importance of this reaction, especially in the remote troposphere, was noted by Archibald et al.[21] based on box model analysis and was built on by Fittschen et al.[22] using data from the remote Cape Verde Observatory. Khan et al.[5] included this reaction in a global model and noted the importance of this reaction with respect to background methanol if the channel (4) forming methanol were significant. Recent studies (Millet et al.[23], Khan et al.[5], and Ferracci et al.[20]) suggest that a large $\phi_{CH3OH}$ would dramatically change methanol levels. Applying a yield of 0% for (4) within a global chemical transport model, Müller et al.[18] find that the discrepancy between modelled and measured atmospheric methanol is significantly exacerbated, owing to the loss of CH$_3$OO through reaction with OH, rather than the self-reaction, which yields CH$_3$OH. Only with a yield of 30% for (4) were Müller et al.[18] able to reconcile measured and modelled methanol.

The existing error bounds on the methanol yield, therefore, leave uncertainties not merely on the magnitude but even on the direction of the impact of this reaction, which highlights the need for direct experimental quantification of the yield. Ferracci et al.[20] argue that the total rate coefficient indicated by the most recent determinations[14,15], lower than that used by Müller et al.[18] would place even more stringent requirements on the methanol yield needed to improve model-measurement agreement; the yield of (4) would need to be in excess of 0.8 to reconcile modelled and measured methanol abundances. We report direct determinations of the methanol yield using two different experimental approaches: isotopologues of OH + CH$_3$OO via multiplexed photoionization mass spectrometry (MPIMS) and a chamber study coupled to proton-transfer reaction time-of-flight mass spectrometry (PTR-TOFMS).

## Results

**Pulsed photolysis MPIMS experiments.** The products of the OH + CH$_3$OO reaction were quantified at 30 Torr in pulsed photolytic experiments using the Sandia multiplexed photoionization mass spectrometer, and at 740 Torr using a new high-pressure reactor, both interfaced with the tuneable-VUV-output of the Chemical Dynamics Beamline (9.0.2) at the Advanced Light Source of Lawrence Berkeley National Laboratory (see the Methods section for further details). At 30 Torr methylperoxy radicals were produced by photolysis of $^{13}$CH$_3$I (to move the methanol mass away from $^{32}$O$_2$ background) in the presence of a large excess of O$_2$, and OH was produced by photolysis of H$_2$O$_2$. In the 740 Torr experiment, reactions of F-atom (generated by

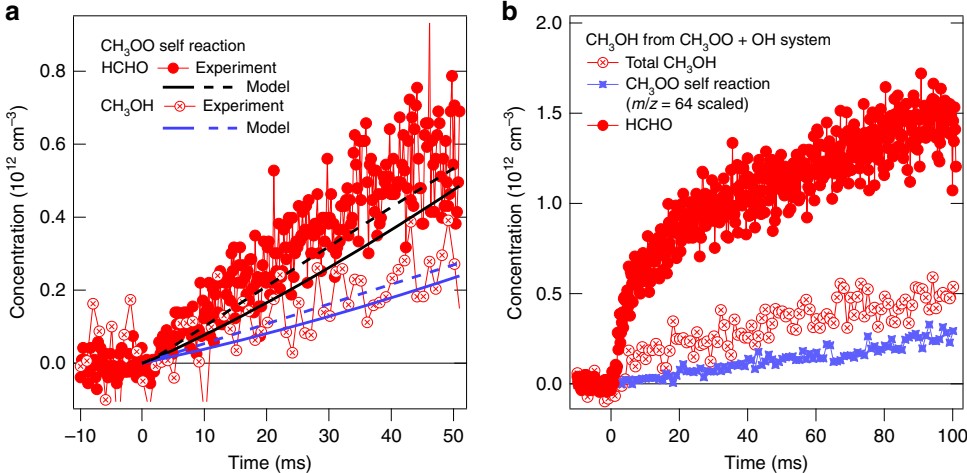

**Fig. 1** Formaldehyde and methanol time profiles from the methylperoxy self- and hydroxyl reactions. Comparison of the contributions from $^{13}CH_3OO$ self-reaction and reaction of $^{13}CH_3OO$ with OH in producing methanol in the photolysis experiments at $P = 30$ Torr. **a** $CH_3OO$ self-reaction (photolysis of $^{13}CH_3I$ in the presence of $O_2$) compared to a kinetic model employing literature rate coefficients and directly measured reactant concentrations, wall loss and two fits to the photolytic depletion. **b** Measurements at the same conditions as (**a**) except with the addition of $H_2O_2$. The contribution from $^{13}CH_3OO$ self-reaction is represented by the signal from another product at $m/z = 64$, $^{13}CH_3OO^{13}CH_3$ (formed only by the self-reaction), scaled using directly measured branching fractions of the self-reaction. The additional, rapidly formed $^{13}CH_3OH$ arises from the reactions of $^{13}CH_3OO$ with OH and $^{13}CH_3O$ with $HO_2$. The temporal resolution of the methanol and $CH_3OOCH_3$ signals is here reduced by a factor of five to more clearly show the amplitudes

photolysis of $XeF_2$)[15] with $CH_4$ and $D_2O$ produced $CH_3$ and OD in the presence of $O_2$. Photoionization mass spectrometry detects precursors, intermediates, and products. Primary and secondary reaction products were observed, including HCHO, $HO_2$ and methanol, confirmed by their photoionization spectra (see Supplementary Figs. 4, 5)[24,25]. Known photoionization cross-sections[24,26] are used to quantify reactant ($H_2O_2$, $D_2O$) and product concentrations in the photoionization measurements (see Supplementary Note 1 and Figures therein). The competition between $CH_3OO$ self-reaction and reaction with OH was assessed through a chemical kinetic model (see Supplementary Note 2, Tables and Figures therein). Figure 1 illustrates the relative contribution of $CH_3OO$ self-reaction to methanol production for representative experiments with and without $H_2O_2$; the data clearly show an additional source of methanol upon addition of $H_2O_2$, which can be attributed to reactions of $CH_3O$ with $HO_2$[27] and branching to channel (4). No evidence is found for the formation of the Criegee intermediate (reaction 3), consistent with the upper limit of 5% reported elsewhere[14,19].

The observed product concentrations were compared to a kinetic model including the $OH + CH_3OO$ reaction, with the branching fraction of $CH_3OH$ from the $OH + CH_3OO$ reaction as a fitted parameter (Supplementary Note 2). Absolute concentration determinations as shown in Fig. 1a display significant sensitivity to the absolute concentration calibration and photolytic depletion. Because the relative photoionization cross-sections of methanol, formaldehyde and $H_2O_2$ are better-known from the measurements of Dodson et al.[24] than are the absolute cross-sections, the most reliable determination of the branching rests on a quantification of the ratio of formaldehyde to methanol. Moreover, because in this reaction system formaldehyde and methanol principally have common sources, the ratio of concentrations normalises for many factors and provides dramatically reduced parametric sensitivity, as can be seen in Supplementary Fig. 3 for the same data set. The dominant uncertainty (see Supplementary Note 4, Tables and Figures therein) is the $\pm 15\%$ uncertainty in the relative cross-sections for formaldehyde and methanol[24], with smaller uncertainties from the rate coefficient for $CH_3O$ with $HO_2$ and for I atom with $HO_2$.

Propagated uncertainties in the total rate coefficient for the reaction of $OH + CH_3OO$, in the absolute concentration calibration, and in the photolysis fraction used to initialise the simulation are insignificant contributors to uncertainty in the branching fraction. We derive a methanol branching fraction of 9 ($\pm 5$)% (assuming negligible branching to (3) and (5)) from a series of six measurements of the $^{13}CH_3OO + OH$ reaction. Figure 2 shows results for a representative measurement. Observations of a small methanol yield strongly support the theoretical value (~7%) for $\varphi_{CH3OH}$ from Müller et al.[18].

The calculations by Müller et al.[18] also showed that at increased pressures, a greater fraction of the trioxide association product (5) is stabilised, predicting a trioxide fraction of approximately 11% at ~1 atm total pressure (c.f. ~0.02% at 30 Torr), at the expense of bimolecular product channels. We find significant evidence for the stabilisation of the trioxide at 740 Torr, but not at total pressures $\leq 30$ Torr, consistent with the calculations of Müller et al. (see Supplementary Note 6 and Figures therein). We are unable to determine an absolute experimental yield for the trioxide, as the photoionization cross-section is unknown. However, assuming that the trioxide photoionization cross-section is comparable to that of methanol would give a yield in the range ~3–12% at 740 Torr (using He bath gas). This is consistent with the 9.6% value calculated for the same pressure (where the bath gas is air) from the expression of Müller et al.[18].

The experiment at 740 Torr is designed so that the ratio of $CH_2O$ to $CH_3OD$ in this system is highly sensitive only to the branching ratio ($k_2/k_4$), and insensitive to photolysis fraction or overall rate coefficients. Analysis returns $\phi_{CH3OH} = (6 \pm 2)$% (see Fig. 2b), indicating, even allowing for a possible kinetic isotope effect, at most a weak negative pressure dependence in the methanol branching fraction, consistent with calculations[18].

**Continuous photolysis chamber experiments.** Reactions were also carried out at atmospheric pressure in a 300 L Teflon bag[28], using different detection techniques (connected to a PTR-TOFMS and an $O_3$ analyser through Teflon tubing) and generation of the

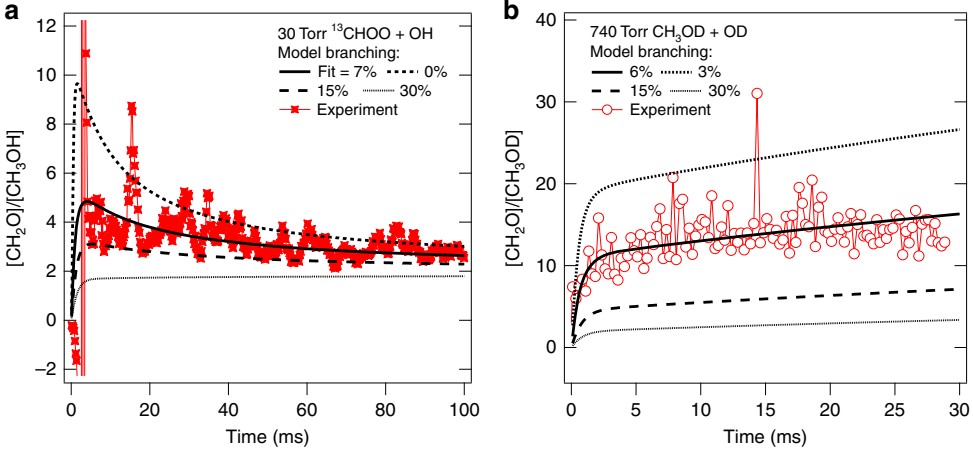

**Fig. 2** Measured and modelled concentration ratios of formaldehyde to methanol. **a** Conditions of Fig. (1b) for the $^{13}CH_3OO + OH$ system modelled using several assumed $CH_3OH$ branching fractions. **b** Products from reaction of OD with $CH_3OO$, initiated by F-atom reaction with $D_2O$ and $CH_4$ in the presence of $O_2$, modelled using several assumed $CH_3OD$ branching fractions. Experimental ratios are taken from 11-point smoothed data to reduce singularities

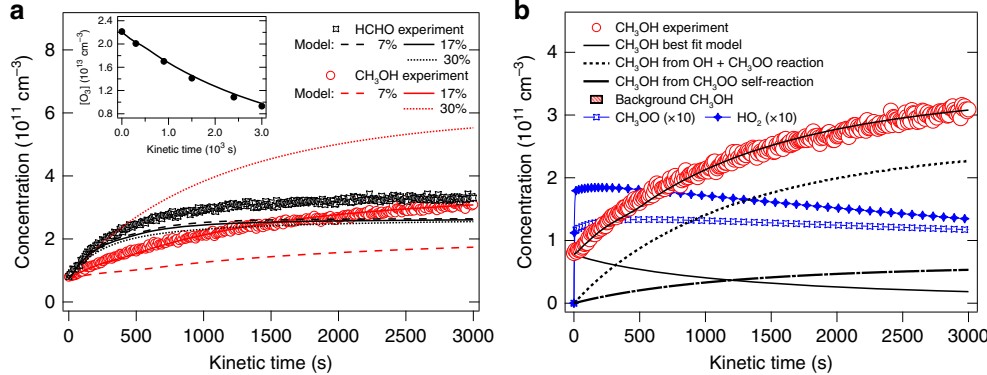

**Fig. 3** Measured and modelled methanol and formaldehyde time profiles. **a** Full lines with model from Supplementary Table 1 (adjusted to $O_3$ decay rate, inset) with $\phi_{CH3OH} = 17\%$, dotted line $\phi_{CH3OH} = 30\%$, dashed line $\phi_{CH3OH} = 7\%$. **b** The solid black line shows the modelled $CH_3OH$ concentration with major contributions being production from self-reaction (dashed-dotted line) and reaction of $CH_3OO + OH$ (dashed line) and removal by the reaction of $CH_3OH$ with OH. $HO_2$ and $CH_3O_2$ concentrations from the model are also shown

reactants. Oxygen ($^1D$, $^3P$) atoms were formed by 254 nm photolysis of $O_3$ in the presence of $CH_4$ and $H_2O$ at 760 Torr of synthetic air, producing OH radicals (further details in Supplementary Note 3). The OH formed $CH_3OO$ through reaction (1), which after a few seconds reached a steady-state concentration (same as $HO_2$, blue open squares and blue solid diamonds, Fig. 3b) at a level where its reaction rate with OH was competitive with the reaction rate of OH with $CH_4$. The profiles in Fig. 3 were modelled (Supplementary Figs. 6, 7) with the $O_3$ photolysis rate and $\varphi_{CH3OH}$ as the only adjustable parameters. Reaction conditions and depletions were chosen such that the reaction of $CH_3OO$ with OH remained the major source for $CH_3OH$ (dashed line in Fig. 3b) with only minor contribution from self-reaction (dashed-dotted line). $CH_3OH$ profiles are very sensitive to the $CH_3OH$ yield in reaction (2), as demonstrated in Fig. 3a. A total of six experiments were carried out with different $O_3/CH_4$ ratios, surface/volume ratios and photolysis rates (Supplementary Table 3); the $CH_3OH$ profiles of all experiments can be reproduced with $\varphi_{CH3OH} = (17 \pm 3)\%$.

The methanol fractions obtained in the chamber experiments are higher than in the pulsed photolysis experiments, even considering the respective error bars. The relevant differences between the experiments lie in the sampling, detection, and residence time. Based on these three factors we conclude that the

stabilised trioxide (5), with a predicted[18] yield of ~11% at atmospheric pressure, and observed in the 740 Torr MPIMS experiments (Supplementary Note 6 and Figures therein), could undergo water-assisted heterogeneous conversion to methanol (a pathway discussed by Müller et al.[18]) in the chamber or sampling line, or fragment upon protonation in the PTR-TOFMS detection system, as has been observed for many organic species[29–31]. The laser photolysis measurements probe reaction times before substantial heterogeneous reaction and directly photoionize molecules sampled by rapid molecular beam expansion. PTR-TOFMS does not detect sizeable concentration of $CH_3OOOH$ (Supplementary Fig. 6) at its parent protonated mass ($CH_5O_3^+$). We have undertaken preliminary ab initio calculations that demonstrate that even if trioxide survives the sampling, it may be detected as protonated methanol (Supplementary Note 3). Two protonation sites are energetically feasible, α and γ with respect to the methyl group. The γ-protonated trioxide is unstable and fragments to $H_2O + CH_3O_2^+$. The α-protonated trioxide is thermodynamically unstable even relative to $^1O_2 + CH_5O^+$. Direct dissociation of the bare cation ($CH_5O_3^+$) has a barrier, but there is a barrierless water-catalysed dissociation pathway (Supplementary Fig. 12). $H_2O$ is present in close proximity to the newly protonated trioxide as a result of the proton-transfer reaction in the PTR-TOFMS detection system and is also present

in appreciable concentrations as a reaction precursor ($2.5$–$3.8 \times 10^{16}$ molecule cm$^{-3}$ and higher in the PTR-TOFMS chamber due to the injection of water to produce $H_3O^+$). It is, therefore, likely that the appreciable yield of trioxide stabilised at the higher pressures of the chamber experiments will lead to additional signal at the protonated methanol mass, resulting in artificially enhanced methanol yields in the chamber experiments. Because there is no method for calibrating the PTR-TOFMS for trioxide, the degree of interference cannot be directly determined.

**Atmospheric model**. To determine the effect of the OH + $CH_3OO$ reaction and its branching on tropospheric composition, we compared a STOCHEM-CRI model that included the title reaction (total rate coefficient from Assaf et al.[15]) and $\varphi_{CH3OH} = 7\%$ to a base case integration that omitted it. This base case scenario is in accordance with other studies[18] that isolate the effect of the OH + $CH_3OO$ reaction; any assignment of $\varphi_{CH3OH}$ would serve to change predicted $CH_3OH$, $HO_x$, or products of $CH_3OO$ reactions. Compared to the base case, addition of the OH + $CH_3OO$ reaction with the $\varphi_{CH3OH} = 7\%$ made only small changes to the global burdens of OH ($-0.9\%$), CO ($+1.0\%$), $O_3$ ($-1.3\%$), $CH_3OH$ ($-1.7\%$) and HCHO ($-0.6\%$) (where the values given in parenthesis are the average values for all grid bases) but had a substantial impact on the global burdens of $HO_2$ ($+7\%$), $CH_3OO$ ($-19.6\%$), $CH_3OOH$ ($-11.7\%$) and other alkyl hydroperoxides (ROOH) ($+4.8$). The OH + $CH_3OO$ reaction decreases $CH_3OO$ ($-19.6\%$) because of removal via reaction with OH and increases the production flux of $HO_2$ through reaction 2 and thereby increases the production of other ROOH. Simultaneously, reaction 4 increases the net production flux of methanol by only 3 Tg/yr from the base case scenario, with 28.7 Tg/yr obtained from peroxy radical reactions (within the range of previous estimates of 15–38 Tg/yr)[18,23].

However, a $\varphi_{CH3OH}$ of 17% (corresponding to the yield from the chamber experiments of this study, uncorrected for the trioxide interference, see Supplementary Fig. 17) increases the global burden of methanol by 14% from the base case scenario, which is lower than the study of Ferracci et al.[20], which found 36% increment of methanol abundances with $\varphi_{CH3OH}$ of 20% from the scenario with $\varphi_{CH3OH}$ of 0%. Under these assumptions methanol production is found to be 54.3 Tg/yr, compared to 116.7 Tg/yr (direct production of 66.1 Tg/yr and indirect production through trioxide formation of 50.6 Tg/yr) estimated for $\varphi_{CH3OH} = 30\%$ by Müller et al.[18].

To reconcile modelled and measured methanol abundances, Müller et al.[18] utilised a yield of 30% for reaction (2c), the upper limit of their calculated range and also the higher rate coefficient[32], $k = 2.8 \times 10^{-10}$ cm$^3$ molecule$^{-1}$ s$^{-1}$. However, Ferracci et al.[20] used $k = 1.6 \times 10^{-10}$ cm$^3$ molecule$^{-1}$ s$^{-1}$ in their modelling study and found comparatively lower $CH_3OH$ production (60 Tg/yr) using the yield of 40%, suggesting that a far higher yield would be needed to reconcile models with measurements. The experimental data presented herein demonstrates that the branching fraction at 298 K is instead closer to the calculated value of ~7% producing only 22.4 Tg/yr methanol, which is smaller than required to rationalise atmospheric observations.

The spatially resolved changes in annual surface levels, compared with the base case integration, are shown in Fig. 4 and discussed in detail in Supplementary Discussion 1. Modest impact is observed on the abundances of OH ($-8\%$), $O_3$ ($-4\%$), CO ($+2.5\%$) and HCHO ($-2.5\%$), and significant changes are observed for $HO_2$ ($+25\%$), $CH_3OOH$ ($-18\%$) and ROOH ($+40\%$) (where the values stated in parenthesis are the maximum changes obtained). Increases in $CH_3OH$ are found over terrestrial

locations, but substantial reductions of up to 30% are estimated in remote tropical regions. Here, the reduction in $CH_3OO$ due to its reaction with OH has retarded the in-situ production of $CH_3OH$ through the self-reaction of $CH_3OO$ and its cross-reactions with other peroxy radicals at remote sites[33]. Therefore, rather than provide a new source of remote $CH_3OH$, with a ~7% yield of 2(c), inclusion of the OH + $CH_3OO$ reaction exacerbates the underestimation of remote $CH_3OH$.

Further integrations with different methanol yields ranging from 0.1 to 1 from OH + $CH_3OO$ (Fig. 5) show that the reaction can be a significant source of methanol over tropical oceans only when $\phi_{CH3OH}$ is higher than 0.15, consistent with other results[18,20], which can be considered to be the compensation point when reaction (4) can begin to contribute to tropospheric $CH_3OH$ over remote tropical oceans. Using the experimentally determined methanol branching fraction of 6%–9% would lead to a significant decrease in atmospheric methanol, specifically in remote regions.

Figure 6 shows a model comparison with a representative data set, also used in earlier comparisons[18], which shows that the reaction 4 has little impact on modelled $CH_3OH$ level over mid-latitudinal remote oceanic areas (e.g., Atlantic, Pacific). Müller et al.[18] use a fraction of 0.65 for tropospheric conversion of trioxide to $CH_3OH$, assuming gas-phase release of methanol from the condensed phase. However a peak fraction of only about 0.2 (centred above the tropical oceans) is predicted to directly produce gas-phase methanol, with most trioxide removed by wet deposition (and condensed phase formation of $CH_3OH$)[18]. Assuming 65% conversion of trioxide to methanol, in addition to our experimentally determined methanol yield of 6–9%, leads to an effective yield around the compensation point of 15%.

## Discussion

The yield of methanol determined experimentally here for the cross-reaction of two important oxidants, OH and $CH_3OO$, agrees with the small methanol production predicted by Müller et al.[18]. In their calculations, the constrained nature of the transition state to hydrogen transfer leads to a preference for direct scission of the pre-product complex to the $HO_2$ and $CH_3O$ products over methanol formation on both the singlet and triplet surfaces. The present results confirm this preference for $CH_3O$, but because both spin manifolds can produce both products, and the uncertainty in the yield encompasses the predicted methanol branching fractions on both surfaces, it is difficult to draw strong conclusions about the role of intersystem crossing in the reaction. However, the experiments considerably improve the uncertainty bounds on the yield, with the MPIMS-determined yields in remarkable agreement with the high-level calculations. Characterisation of the methanol signal in the MPIMS experiment via photoionization energy spectroscopy through comparison with the known literature photoionization energy spectra (Supplementary Figs. 4, 5) and cross-sections shows that the methanol branching fraction can be robustly determined. Our calculations (Supplementary Note 3) demonstrate the potential contribution of the trioxide to the methanol signal in the PTR-TOFMS detection, in addition to possible heterogenous conversion pathways. This is consistent with our detection of the trioxide at higher pressures in the MPIMS experiment. The MPIMS value of 6–9% reflects the direct reaction product branching fraction.

Inclusion of this reaction in a global atmospheric chemistry and transport model could not improve the methanol discrepancy between model and observations; the direct methanol branching fraction results in a factor of 1.5 underprediction of methanol in

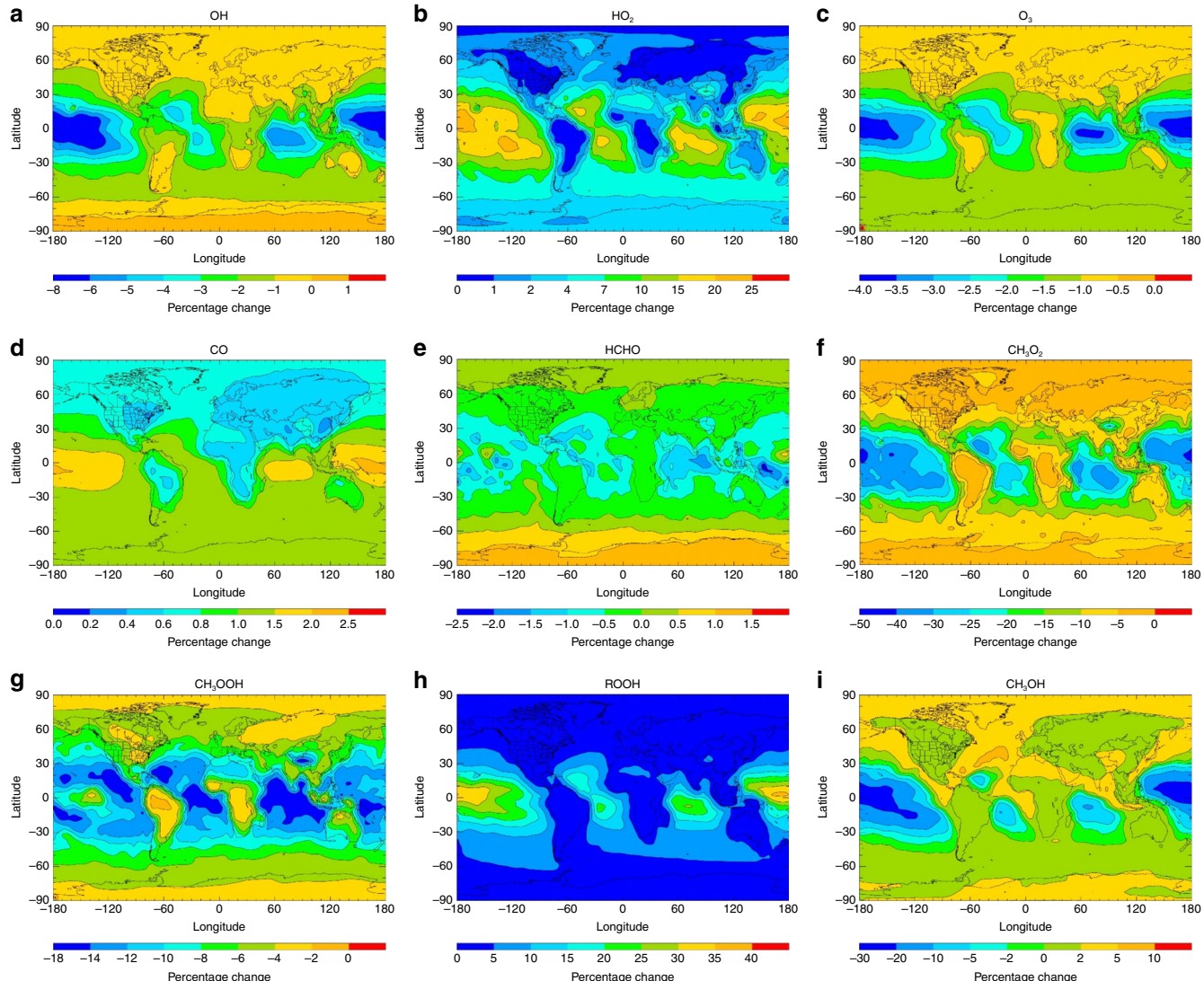

**Fig. 4** The impact of the title reaction on key atmospheric species with a 7% methanol yield. Annual surface percentage changes in **a** OH **b** HO$_2$ **c** O$_3$ **d** CO **e** HCHO **f** CH$_3$O$_2$ **g** CH$_3$OOH **h** ROOH (excluding CH$_3$OOH), and **i** CH$_3$OH upon inclusion of the OH + CH$_3$OO reaction with assumed branching fractions $\varphi_2 = 0.93$, $\varphi_3 = 0.00$, $\varphi_4 = 0.07$, $\varphi_5 = 0.00$

remote environments. This work highlights the necessity for further characterisation of potential atmospheric methanol sources, including understanding the tropospheric fate of the trioxide (5). Moreover, Khan et al.[3] determined that up to 17% of peroxy radicals may be complexed to a single water molecule under atmospheric conditions, and previous work has demonstrated an impact of water complexation on reaction rate coefficients and product branching fractions[34,35]. The near-atmospheric pressure measurements here were carried out at low relative humidity (RH); further investigations as a function of RH may help to determine whether a water effect on reaction 2, or perhaps unexplored functionalized peroxy radical cross-reactions, may be part of the missing source of atmospheric methanol.

## Methods

**Experiments**. Measurements were performed at 298 K and 30 Torr using the Sandia Multiplexed Photoionization Mass Spectrometer (MPIMS) instrument coupled to the tuneable-VUV-output of the Chemical Dynamics Beamline (9.0.2) at the Advanced Light Source, Lawrence Berkeley National Laboratory. Reagent (CH$_3$I or $^{13}$CH$_3$I, H$_2$O$_2$, O$_2$) and bath gases (He) were flowed into a halocarbon wax-coated quartz reactor via a set of calibrated mass flow controllers. H$_2$O$_2$ was produced by heating urea hydrogen peroxide and was entrained into the He flow

via a pressure- and temperature-controlled bubbler. At the high concentration of O$_2$ utilised in the experiments ($2.6 \times 10^{17}$ cm$^{-3}$) a significant O$_2$ peak was observed $\sim m/z = 32$, ionised by the small amount of transmitted higher undulator harmonics. Because the masses of $^{16}$O$_2$ (31.98984 amu) and $^{12}$CH$_3$OH (32.02622 amu) could not be completely resolved, experiments were performed using $^{13}$CH$_3$OO, such that the resultant methanol signal was well separated in mass from O$_2$. OH and $^{13}$CH$_3$, the chemical precursor to $^{13}$CH$_3$OO, were produced photolytically via a 248 nm excimer laser aligned along the axis of the reactor. $^{13}$CH$_3$OO was produced from the subsequent reaction of $^{13}$CH$_3$ with excess O$_2$, yielding a [$^{13}$CH$_3$OO] excess over [OH] of a factor of 3–6. Reactant and product species were continuously sampled via a 600 μm orifice in the reactor sidewall. The resultant molecular beam was intercepted by the ionising tuneable-VUV radiation, and ions were detected via time-of-flight mass spectrometry. Single photon-energy measurements at 11 eV yielded simultaneous obtained kinetic plots over the whole m/z range (~2–159 amu). Photoionization spectra, whereby the ionisation energy was scanned stepwise in 25 meV steps, are used to definitively identify the detected species.

Additional measurements performed at 740 Torr were carried out in a quartz-lined metal reactor sampled through the end wall. The higher-pressure experiments produced CH$_3$ and OD by reaction of F atoms (formed by 248 nm photolysis of XeF$_2$) with CH$_4$ and D$_2$O in the presence of ~$10^{18}$ cm$^{-3}$ [O$_2$] and excess He. Details of the conditions at which the experiments were performed, and the chemical model are given in Supplementary Notes 2 and 4.

Chamber experiments were performed in a 300 L Teflon simulation chamber suspended in a closed box where photolysis of O$_3$ in the presence of water vapour ($2.5$–$3.8 \times 10^{16}$ cm$^{-3}$) was carried out using 1–8 Hg lamps. Methane (5% in N$_2$, Air

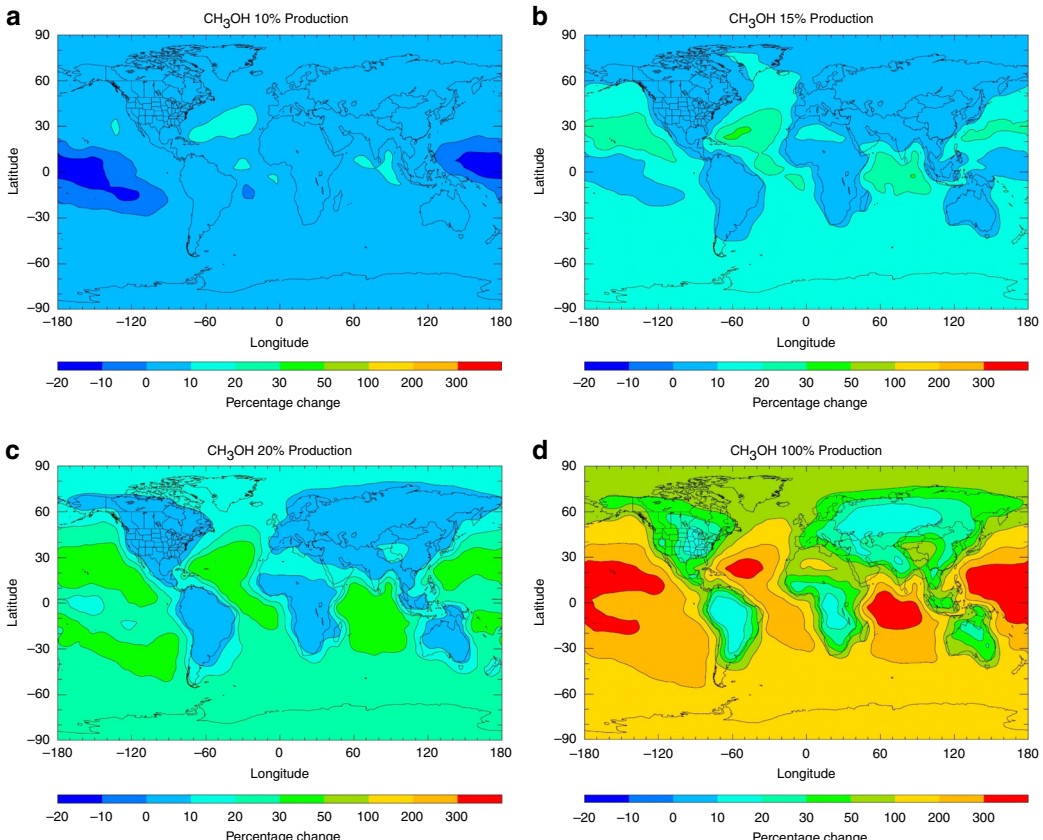

**Fig. 5** The impact of the title reaction with different methanol yields on global methanol. Annual surface percentage changes in $CH_3OH$ on inclusion of the $OH + CH_3OO$ reaction compared with the base case model with the branching fractions for channel 4 of **a** 0.1 **b** 0.15 **c** 0.2 and **d** 1 and for channel 2 of **a** 0.9 **b** 0.85 **c** 0.8 and **d** 0, respectively

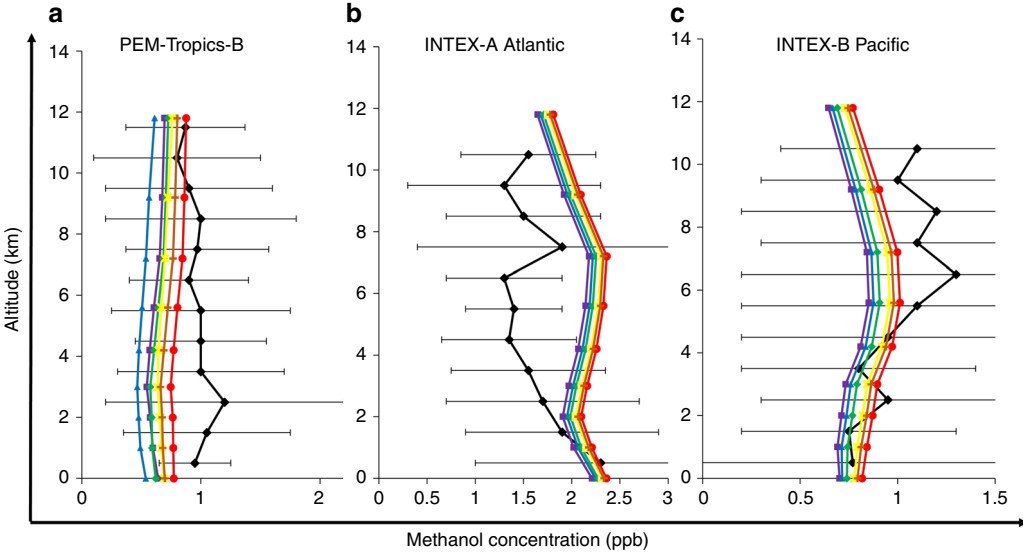

**Fig. 6** Comparison of methanol field measurements with modelled outcomes as a function of altitude. Vertical profiles of measured and modelled $CH_3OH$ over **a** tropical Pacific **b** midlatitude Atlantic and **c** Pacific. The data compilation of Müller et al.[18] containing measured $CH_3OH$ are used for the model-measurement comparison. Violet square symbols represent mean $CH_3OH$ produced for the base case (without the $OH + CH_3OO$ reaction). Blue triangle, green diamond, yellow star, orange plus and red circle symbols represent mean $CH_3OH$ produced at the branching fractions of the channel of 0.07, 0.1, 0.15, 0.17 and 0.2, respectively. Black triangles represent the measurement $CH_3OH$ data and the black error bars represent measurement variability (standard deviation)

Liquide) was introduced in the reactor using ml syringes to get initial concentrations between $1.8 \times 10^{14}$ and $3.7 \times 10^{14}$ cm$^{-3}$. The CH$_4$ relative concentrations were determined both by infrared spectroscopy at around 1800 cm$^{-1}$ and by high-resolution proton-transfer reaction time-of-flight mass spectrometry (HR-PTR-TOFMS Ionicon 8000). Although the CH$_4$ proton affinity is lower than the water proton affinity, a small signal scaling with the CH$_4$ concentration was detected by the PTR-TOFMS instrument due to the high concentrations used in these experiments. Ozone was produced by a commercial O$_3$ generator (C-Lasky, AirTree Europe GmbH) and initial concentrations of $(2.0–8.4) \times 10^{13}$ cm$^{-3}$ were obtained. The O$_3$ time-dependent concentrations were measured using a UV-absorption analyser (Environnement SA 42 M) while methanol and formaldehyde were measured by PTR-TOFMS. Absolute methanol concentrations were determined after daily calibrations of the PTR-TOFMS transmission curve using a Gas Calibration Unit (GCU, IONICON) and a gas standard composed of methanol, acetaldehyde, acetone, benzene, toluene, o-xylene and 1,2-dichlorobenzene (IONICON, 1σ uncertainty for each species of 5–6%). Formaldehyde measurements were also calibrated by adsorption on 2,4-DiNitro Phenyl Hydrazine cartridges and analysis through High-Pressure Liquid Chromatography (HPLC-UV) for some of the experiments[36]. The agreement between PTR-TOFMS and HPLC was within 20%. Methanol and formaldehyde wall losses were also negligible (<1% h$^{-1}$).

**Model**. The STOCHEM-CRI model has been described in previous papers[3,5,37,38] and pertinent details are given herein.

STOCHEM is a global 3-dimensional chemistry transport model that adopts a Lagrangian approach splitting the troposphere into 50,000 constant mass air parcels. As it is a Lagrangian model, the transport and chemistry can be decoupled and hence these air parcels are advected with a 3-hour time step by meteorological data from the UKMO Hadley Centre global general circulation model called the Unified Model (UM)[39]. The UM works on a grid resolution of 1.25° longitude, 0.833° latitude and 12 unevenly spaced (with respect to altitude) vertical levels between the surface and a upper boundary of ~100 mb[40]. The description about the meteorological parameterisations (e.g., vertical coordinate, advection scheme, boundary layer treatment, inter-parcel exchange and convective mixing) can be found in Percival et al.[41]. The model used in this experiment is an 'offline' model and hence the meteorological data are archived within the code itself. Each air parcel contains the complete 229 species in the code, which can take part in any of the reactions detailed. The chemical mechanism used in STOCHEM is the common representative intermediates version 2 and reduction 5 (CRI v2-R5), which was built using a series of five-day box model simulations on each compound, on a compound-by-compound basis. The performance of the chemistry of these simulations was optimised using the Master Chemical Models (MCM version 3.1)[42]. More details of the CRI v2-R5 mechanism can be found in Watson et al.[43] and Utembe et al.[37,42]. The photolysis reactions were calculated by integrating (overall wavelengths) the product of flux, absorption cross-section and quantum yield[40], which were included in the model as described in Khan et al.[3]. In addition to the gas-phase chemical reaction and photolysis, the air parcels also consider emissions and physical removal processes (dry and wet deposition). Air parcels within the planetary boundary layer can see removal of certain species by dry or wet deposition. The rate of dry deposition is dependent on whether the air parcel is over land or ocean with appropriate species dependent deposition velocities. The dry deposition velocities used in STOCHEM were adapted from the annual mean values calculated using the MATCH global model[45]. The scavenging coefficients for convective and dynamic precipitation taken from Penner et al.[46] were combined with precipitation rates and scavenging profiles to calculate the loss rates of species (wet deposition) from an air parcel. Emissions are treated as an additional term to the source fluxes of each species during each integration time step, rather than a step change in species concentration[40,47]. Emissions are from the surface and split into five main classes: anthropogenic, biomass burning, oceans, soils and vegetation. Emission data is mapped onto a monthly 5° longitude × 5° latitude resolution, two-dimensional source map. The emissions data employed in the base case STOCHEM model were adapted from the Precursor of Ozone and their Effects in the Troposphere (POET) inventory for the year 1998[48]. More details about the emissions data can be found in Khan et al.[5]. The concentrations produced from an integration is mapped onto an Eulerian grid resolution 5° × 5° with 9 vertically spaced pressure levels, each 100 hPa thick. Summing the 50,000 air parcels produces a global burden for each species, which can be broken down into the respective source and sink fluxes.

The flux outputs are calculated within each grid square by dividing the average flux per air parcel by its volume, which gives volume-averaged fluxes with units of cm$^{-3}$ s$^{-1}$. In this study, base case experiment involves the STOCHEM being integrated with the CRI v2-R5 mechanism subsequently referred to as 'STOCHEM-base'[3,5,37]. A further simulation was performed in the study, which involved the STOCHEM-base being integrated after including the reaction OH + CH$_3$OO with two product channels (2 and 4). The simulations were conducted with meteorology from 1998 for a period of 24 months with the first 12 allowing the model to spin up. Analysis were performed on the subsequent 12 months of data.

## Data availability

The datasets generated in the current study are available from the corresponding authors on reasonable request.

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

## Acknowledgements

This material is based upon work supported by the Division of Chemical Sciences, Geosciences and Biosciences, Office of Basic Energy Sciences (BES), U.S. Department of Energy (USDOE). Sandia National Laboratories is a multimission laboratory managed and operated by National Technology and Engineering Solutions of Sandia, LLC, a wholly owned subsidiary of Honeywell International, Inc., for the USDOE's National Nuclear Security Administration under contract DE-NA0003525. This paper describes objective technical results and analysis. Any subjective views or opinions that might be expressed in the paper do not necessarily represent the views of the USDOE or the United States Government. The development and operation of the high-pressure kinetic photoionization apparatus was supported by the Division of Chemical Sciences, Geosciences and Biosciences, BES/USDOE, through the Argonne-Sandia Consortium on High Pressure Combustion Chemistry. The Advanced Light Source is supported by the Director, Office of Science, BES/USDOE under Contract DE- AC02-05CH11231 at Lawrence Berkeley National Laboratory. This Research was carried out in part by the Jet Propulsion Laboratory, California Institute of Technology, under contract with the National Aeronautics and Space Administration (NASA), supported by the Upper Atmosphere Research and Tropospheric Chemistry programs. AK and DES were funded by NERC (NE/K004905/1). SAGE and PC2A laboratory acknowledge funding by the French ANR agency under contract No. ANR-11-LabX-0005-01 CaPPA (Chemical and Physical Properties of the Atmosphere), the Région Hauts-de-France, the Ministère de l'Enseignement Supérieur et de la Recherche (CPER Climibio) and the European Fund for Regional Economic Development. A. Grira is grateful for a PhD grant from Brittany Region and IMT Lille Douai.

## Author contributions

R.L.C., L.S., C.F., A.T., S.D., C.J.P., D.E.S. and C.A.T. planned the experiments. R.L.C., M.A.H.K., J.Z., D.E.S. and C.A.T. carried out the modelling work. Data analysis was undertaken by R.L.C., L.S., C.F., C.S., S.D., A.T. and C.A.T. Experimental work was performed by R.L.C., L.S., I.O.A., B.R., K.R., K.A., M.W.C., D.R., D.L.O., M.D., A.G., C.J.P. and C.A.T. The manuscript was prepared by R.L.C., M.A.H.K., J.Z., L.S., I.O.A., B.R., K.R., M.W.C., D.R., D.L.O., C.F., C.S., S.D., A.T., C.J.P., D.E.S. and C.A.T.

## Additional information

**Competing interests:** The authors declare no competing interests.

