## [Peer Review File · Nature Communications]

Reviewers' comments:

Reviewer #1 (Remarks to the Author):

The authors present new experimental results quantifying the methanol yield from $\text{CH}_3\text{OO} + \text{OH}$, which is a reaction that has received recent attention in atmospheric chemistry as a more important pathway than previously realized. Prior work had postulated that a high methanol yield from this reaction could reconcile model underpredictions of methanol in the remote atmosphere. Here the authors show that the methanol yield is in fact lower than previously assumed in those studies, and too low to account for the so-called missing atmospheric methanol. The results thus shed new light on an important atmospheric reaction and have implications for atmospheric modeling.

One reservation I have with this work as presented is the following. The authors present two experimental estimates of the yield: 6-7% and 17%, but then the entire narrative and conclusions are constructed only in terms of the former result. For example, in the abstract: "a branching fraction below 15% is established"; page 9: "the experimental data demonstrate that the branching fraction is closer to 7%"; page 11: "the experimentally determined methanol branching fraction of 6% - 9%". The authors speculate that the 17% value may be affected by an experimental artifact (heterogeneous conversion of CH_3OOOH to CH_3OH), but this is just speculation and not sufficiently convincing to disregard the 17% result in the uncertainty range. The authors should either provide more robust evidence that the 6-7% result is the one to be believed or else reconstruct their uncertainty ranges and interpretation / analysis to encompass the 17% value.

Secondly, a few parts of the paper are somewhat sloppy or misleading in the way results are presented; however, this is easily corrected with minor changes. More specific comments and suggestions follow (in order of appearance in the manuscript).

P2, 2nd and 3rd paragraphs: there are some more recent global methanol studies that should also be referenced here.

P2, "implicated in secondary aerosol". This is too much of a stretch. The paper cited (which discusses reactions of Criegee intermediates with alcohols) does not claim an important SOA source from methanol at all, but speculates that Criegee intermediate reactions with larger alcohols might have a "subtle" role to play in SOA formation.

P2, "methanol chemistry is connected to initial organic species encompassing both biogenic and anthropogenic emissions as it is an oxidation product of many species, and thus is a benchmark for the performance of atmospheric models". This is misleading. Methanol sources are dominated by direct emissions. What secondary production there is comes about through reactions of CH_3OO which as the authors have already pointed out is mostly from methane. So to say that methanol is a tracer for VOC photochemistry is wrong.

P3, "on THE reaction potential energy surface"

Fig 1 caption, "down-sampled by a factor of five", please clarify what is meant here.

P7, "PTR-TOFMS does not detect sizable concentrations of CH_3OOOH (see Figure S6)". But would CH_3OOOH actually appear in the PTR-MS at m/z 65? It seems likely to fragment.

P8, other alkyl hydroperoxides are found to increase with the addition of the $\text{CH}_3\text{OO} + \text{OH}$ reaction. Is this because of the main channel yielding $\text{CH}_3\text{O} + \text{HO}_2$, so you increase HO_2 , and thereby increase production of other ROOH? Would other ROO also be expected to react rapidly with OH?

P8, what is the fate of the trioxide in the model simulation? If you argue that it converts to CH₃OH in your inlet systems, would the same occur in the atmosphere?

P9, It is not very convincing to state that the "the experimental data demonstrate that the branching fraction is closer to 7%", when you have two results: 7% and 17%. You have speculated that 17% is too high because of trioxide conversion in the experimental apparatus, but that is just speculation and without further evidence not sufficient to exclude 17% from your uncertainty range. It is fair to state that your best estimate is 7%, but the uncertainty range should encompass the other experimental results.

The phrasing in this section as to what is or is not a 'substantial' source of methanol is a bit misleading. Global land emissions of methanol are somewhere around 100 Tg/y and atmospheric production from peroxy radical cross and self reactions has been estimated at ~40 Tg/y. So even with a methanol yield of only 7% the CH₃OO + OH reaction still provides a significant augmentation (22 Tg) to the known atmospheric source of methanol, as does your upper estimate of 17% (54 Tg). It is just that both are smaller than estimated by Muller, and smaller than needed to explain atmospheric observations.

Fig. 6: "black error bars represent measurement variability". Say what that actually means. SD?

P11, "experimentally determined branching fraction of 6-9%". Again, the two experimental values are 7 and 17%. You haven't convincingly discredited the 17% value.

P10, "Modest changes are observed on the abundances of OH (-8%) ..."; for all the percentages listed in this section, the wording needs to specify that the percentages listed are "up to" or "maximum", otherwise it gives a misleading impression. -8% would be quite a large global average OH change, but in fact that is the maximum value seen.

P11, "using the experimentally determined methanol branching fraction of 6-9% would lead to a significant decrease in atmospheric methanol". Earlier you said that a 7% yield increases the global methanol production rate by 3 Tg/y, so this part needs to be clearer about what you mean. Did you mean relative to the higher-yield scenarios, or specifically in remote regions?

Fig 6 would be easier to read if the colored lines had an intuitive color progression from low to high yield, e.g., from cool to warm colors.

P15 of supplement: "a slight increment (1%) in the southern hemisphere ocean regions where NO_x is high in the model" ... this seems odd to have high NO_x in remote southern ocean areas. Please state why, is it due to fires?

P15 of supplement, "driven by to major NO₂ formation reactions, CH₃OO + NO (22%) and HO₂ + NO (56%)" ... It is unclear what these percentages represent.

P15 of supplement. For the discussion of changes to the HCHO budget, it is worth also pointing out that any CH₃OH produced will also go on to produce HCHO (aside from minor deposition losses)... in other words, we don't expect very much change to the overall HCHO source regardless of the branching in these reactions.

Reviewer #2 (Remarks to the Author):

This is experimental study of the title reaction confirming the results of a recent theoretical study (ref 15). The experiments appear to be carefully done and well explained. The presentation seems

long for a communication and I would recommend publication as a full paper. Atmospheric model incl. Figure 4-6 – could be reduced without loss of essence.

Minor

Page 2, based on recent results by Brandt et al (Angew Chem 2018, 57, 3820) I would suspect the $\text{CH}_3\text{OO} + \text{RO}_2$ reaction to be slow compared to NO and HO₂ reactions.

Page 3, explain the phi

Page 6, the $\sim 1\text{atm}$, should be reworded. E.g moved till after 11% and say at 1 atm total pressure.

Page 7/8 I find the Results () a bit odd sections.

Page 11, black triangles are hard to see.

Page 12, remove 'difficult stationary points' – what is meant by this

Figure S3, a bit off with blue dash and black line as model, perhaps models

Figure S10, in all panels it seems like the model rise is not levelling off like the expt seem to do.

We would like to thank both reviewers for their careful, instructive and thoughtful comments. After considering their comments, we made substantial clarifications and significant improvements to our manuscript.

The main concern of Reviewer #1 regards the difference in the obtained methanol branching fraction in the two MPIMS experiments (6-7%) compared with the overall methanol yield in the chamber/PTR-TOFMS experiment (17%). Our conclusion after examining possible sources of error in the two methods was that this difference arises from contributions of the trioxide. To clarify and further substantiate this interpretation, we have followed Reviewer #1's suggestions in: (1) gathering new experimental data and undertaking *ab initio* calculations to provide evidence that trioxide decomposition in the chamber/PTR-TOFMS experiment leads to enhanced methanol signal, and (2) extending our atmospheric modelling to examine the impact on global methanol were the effective methanol yield from the OH + CH₃OO reaction equal to 17%.

We will here briefly summarize the new work we have undertaken, and the conclusions we draw from this. Further discussion and details can be found within the reviewer replies at each pertinent comment. Due to space limitations in the manuscript, we have referred to our new work in the main paper, but most of the details will appear in the SOM.

- (1) We have recorded new experimental data in the MPIMS experiment at 10, 30 and 740 Torr to find evidence for the formation of the trioxide product. We observe the trioxide only at 740 Torr (as anticipated by the predicted minimal yields at lower pressures), confirming that the trioxide is indeed formed at higher pressures, pertinent to the troposphere, and, by extension, in the chamber/PTR-TOFMS experiment.
- (2) We have undertaken *ab initio* calculations to investigate the fragmentation of the protonated trioxide, as would be detected in the chamber/PTR-TOFMS experiment. We find that protonation can occur at two sites of trioxide. In the presence of water (H₃O⁺ is the proton source), one of these isomers will decompose to yield protonated methanol, whereby the methanol signal observed is artificially enhanced by the contributions from the trioxide.
- (3) We have run our atmospheric model with a 17% yield of methanol. The new data strengthens the demonstration that the 17% overall yield from the chamber/PTR-TOFMS experiment is subject to interference from the trioxide, and that the 6-9% yield from the MPIMS experiments reflect the direct product branching. Nevertheless, because the atmospheric fate of the trioxide may include release of methanol, we consider the potential atmospheric implications of the 17% yield. As shown in the original paper for a 15% yield, a 17% yield provides a small improvement in the agreement between measured and modelled methanol over the remote troposphere compared to the base case, but remains insufficient to reconcile the two.

Specific replies to reviewer comments are given in red below.

Reviewer #1 (Remarks to the Author):

The authors present new experimental results quantifying the methanol yield from $\text{CH}_3\text{OO} + \text{OH}$, which is a reaction that has received recent attention in atmospheric chemistry as a more important pathway than previously realized. Prior work had postulated that a high methanol yield from this reaction could reconcile model underpredictions of methanol in the remote atmosphere. Here the authors show that the methanol yield is in fact lower than previously assumed in those studies, and too low to account for the so-called missing atmospheric methanol. The results thus shed new light on an important atmospheric reaction and have implications for atmospheric modeling.

One reservation I have with this work as presented is the following. The authors present two experimental estimates of the yield: 6-7% and 17%, but then the entire narrative and conclusions are constructed only in terms of the former result. For example, in the abstract: “a branching fraction below 15% is established”; page 9: “the experimental data demonstrate that the branching fraction is closer to 7%”; page 11: “the experimentally determined methanol branching fraction of 6% - 9%”. The authors speculate that the 17% value may be affected by an experimental artifact (heterogeneous conversion of CH_3OOOH to CH_3OH), but this is just speculation and not sufficiently convincing to disregard the 17% result in the uncertainty range. The authors should either provide more robust evidence that the 6-7% result is the one to be believed or else reconstruct their uncertainty ranges and interpretation / analysis to encompass the 17% value.

We are grateful to Reviewer #1 for the careful consideration of our manuscript and the insightful suggestions. We have carried out new experimental, theoretical and modelling work to address their concerns. This new work provides significant evidence that the chamber/PTR-TOFMS experiments can observe contributions from the trioxide product at the mass of the methanol signal, and that the lower yield obtained in the MPIMS experiments reflects the direct branching ratio in the reaction. Responses to specific comments below have more detailed information regarding this new data.

Secondly, a few parts of the paper are somewhat sloppy or misleading in the way results are presented; however, this is easily corrected with minor changes. More specific comments and suggestions follow (in order of appearance in the manuscript).

P2, 2nd and 3rd paragraphs: there are some more recent global methanol studies that should also be referenced here. Thank you for drawing our attention to this. We have added three further references from more recent studies (Millet *et al.* 2008, Stavrakou *et al.* 2011, and Cady-Pereira *et al.* 2012). The Ferracci *et al.* paper has also been referred to earlier than before, with regards to the impact of the OH + CH₃OO on modelled methanol abundances.

P2, “implicated in secondary aerosol”. This is too much of a stretch. The paper cited (which discusses reactions of Criegee intermediates with alcohols) does not claim an important SOA source from methanol at all, but speculates that Criegee intermediate reactions with larger alcohols might have a “subtle” role to play in SOA formation. We have amended our statement to clarify that reactions of alcohols may have subtle indirect effects of secondary organic aerosol formation.

P2, “methanol chemistry is connected to initial organic species encompassing both biogenic and anthropogenic emissions as it is an oxidation product of many species, and thus is a benchmark for the performance of atmospheric models”. This is misleading. Methanol sources are dominated by direct emissions. What secondary production there is comes about through reactions of CH₃OO which as the authors have already pointed out is mostly from methane. So to say that methanol is a tracer for VOC photochemistry is wrong. In the model, ~74% of CH₃OO originates from methane oxidation, whilst the remaining 26% is from the oxidation of other organic species. Whilst, as the reviewer states, direct emissions and methane oxidation dominate, we maintain it is not incorrect to use methanol as a marker for VOC photo-oxidation chemistry. We have rephrased this sentence to emphasize that direct emissions dominate.

P3, “on THE reaction potential energy surface. We have corrected this.

Fig 1 caption, “down-sampled by a factor of five”, please clarify what is meant here. We have amended the caption to state that the signals are re-binned, reducing the temporal resolution by a factor of five.

P7, “PTR-TOFMS does not detect sizable concentrations of CH₃OOH (see Figure S6)”. But would CH₃OOH actually appear in the PTR-MS at m/z 65? It seems likely to fragment. We thank the reviewer for this astute comment identifying another pathway for contribution of trioxide in the PTR/MS experiments. We have undertaken *ab initio* calculations to explore the stability of the protonated trioxide. Three protonation sites have been considered, (on each of the oxygen atoms), α , β and γ with respect to the methyl group. According to the work of Müller *et al.*, only the singlet electronic state of the trioxide is pertinent, and so our calculations have only considered the singlet closed shell electronic state of the trioxide.

Geometry optimization and energy calculations have been performed at the M06-2X/6-311++G** level of theory. The α and β protonated trioxides are both found to be stable. Trioxide which is protonated at the γ site is found to yield a van der Waals-like complex between H₂O and CH₃O₂⁺ fragments. The likelihood of protonation at each of these sites has been investigated through

calculations of the proton affinity of each isomer with respect to water as H_3O^+ is the proton source in the PTR experiments. Protonation at the β site is found to be endothermic with respect to water by $9.4 \text{ kcal mol}^{-1}$, whilst protonation at both the α and γ sites are determined to be exothermic relative to water. As protonation of the trioxide is anticipated to be a barrierless processes, the branching fractions for the protonation sites are likely to be determined by long-range interactions, and so branching fractions for the α and γ sites cannot be readily obtained without rigorous dynamics calculations. We can however conclude the following: some of the trioxide formed in the chamber experiments will likely be protonated at the γ site, leading to an unstable species that will decompose to water + CH_3O_2^+ . We note that the latter has been detected in the present experiments (see SOM, Supplemental Figure 6). Some of the trioxide is also anticipated to be protonated at the α site, which will lead to an apparently stable protonated species.

However, due to the role of protonated water in the proton transfer process and, additionally, the presence of appreciable concentrations of H_2O in the chamber experiments ($2.5\text{-}3.8 \times 10^{16} \text{ molecule cm}^{-3}$, and higher in the PTR-TOFMS chamber due to the injection of water to produce H_3O^+) it is necessary to consider the potential role of water reactions with the α -protonated trioxide. We find a highly exothermic process with a submerged barrier (Supplemental Figure 12) for the decomposition of $[\text{CH}_3\text{OHOOH}\dots\text{H}_2\text{O}]^+$ (formed either during the proton transfer process, or through interaction of water with the α -protonated trioxide, in both cases with considerable internal energy, see Supplemental Figure 12), leading to $\text{CH}_5\text{O}^+ + \text{H}_2\text{O} + \text{O}_2$.

Supplemental Figure 12 ZPE-inclusive stationary point energies for the decomposition of $[\text{CH}_3\text{OHOOH}\dots\text{H}_2\text{O}]^+$ obtained at the M06-2x/6-311++G** level of theory.

The calculations therefore suggest that any trioxide which survives heterogeneous loss would be detected in the PTR system as either CH_3O_2^+ (from the γ -protonated trioxide) or protonated methanol (from the α -protonated trioxide). This is consistent with our observation of larger overall methanol yields obtained in the PTR-TOFMS chamber experiments than in the low pressure and ambient pressure MPIMS experiments.

Unfortunately, we cannot establish a branching fraction for trioxide in the $\text{OH} + \text{CH}_3\text{OO}$ reaction in the chamber experiment using this interference because (1) the branching between α and γ -

protonated trioxide is unknown and (2) the detection efficiency of the protonated trioxide fragments is also not known. However, the calculations of Müller *et al.*, which have proved highly accurate for direct methanol yield, predict 11% trioxide formation.

P8, other alkyl hydroperoxides are found to increase with the addition of the CH₃OO + OH reaction. Is this because of the main channel yielding CH₃O + HO₂, so you increase HO₂, and thereby increase production of other ROOH? Would other ROO also be expected to react rapidly with OH? The reviewer is correct. We have amended the text to clarify this. With regards to the reaction of other ROO + OH reactions – there have been limited kinetic studies on these reactions, but for straight chain alkyl peroxy radicals + OH (C₂H₅O₂, C₃H₇O₂ and C₄H₉O₂), rate coefficients in the range 1.3-1.5 x 10⁻¹⁰ molecule⁻¹ cm³ s⁻¹ have been measured by Assaf *et al.* but dependence on the functional group has not been investigated. The tropospheric impact of these and other OH + ROO reactions is beyond the scope of the present work but should be the focus of future studies. (Assaf *et al.*, *Chemical Physics Letters*, **684**, 245-249 (2017).)

P8, what is the fate of the trioxide in the model simulation? If you argue that it converts to CH₃OH in your inlet systems, would the same occur in the atmosphere? The fate of the trioxide has not been specifically accounted for in the model simulations, however this has been discussed by Müller *et al.* They conclude that the most significant sinks of trioxide in the atmosphere (in order of importance) would be (1) uptake into aqueous aerosols (2) reaction with OH and (3) reaction with water dimer. In the aqueous phase of the aerosol, trioxide is anticipated to decompose into CH₃OH + O₂. Similarly, reaction with water dimer is expected to yield CH₃OH + O₂ + 2H₂O, whilst reaction with OH is thought to lead to formation of CH₃O + O₂ + H₂O. Müller *et al.* postulate that on average, 65% of the stabilized trioxide, formed with a branching fraction of 10.7 % at ambient pressure and temperature, will decompose to methanol (accounting for the above three processes, and assuming the gas-phase release of methanol from the condensed phase) in the atmosphere – therefore contributing an effective ~7.1% (i.e./10.7% x 65%) additional methanol from the reaction of OH + CH₃OO.

Adding this value (7.1%) to our experimentally determined methanol yield from the MPIMS experiments of 6-7% gives ~<15% total methanol yield. The implications of a 15% methanol yield have been modelled in our original submission. This methanol yield represents a 'break-even' point in the remote troposphere, where the amount of methanol produced compensates for how much methanol would have been produced should CH₃OO undergo self- and cross-peroxy radical reactions, rather than reaction with OH. However it does not provide a significant source of tropospheric methanol in the remote regions where the disagreement between modelled and measured values is of most interest.

The reviewer is indeed correct that the eventual fate of the trioxide in the atmosphere may affect tropospheric methanol. However, this manuscript addresses the direct implications of the OH + CH₃OO reaction, as it has been suggested that the direct yield of methanol, as a bimolecular product of this reaction, can rectify the underprediction of tropospheric methanol. The chain of subsequent reactions of the products (such as the trioxide) are potentially interesting for tropospheric modeling but not representative of the fundamental reaction which is being studied.

P9, It is not very convincing to state that the “the experimental data demonstrate that the branching fraction is closer to 7%”, when you have two results: 7% and 17%. You have speculated that 17% is too high because of trioxide conversion in the experimental apparatus, but that is just speculation and without further evidence not sufficient to exclude 17% from your uncertainty range. It is fair to state that your best estimate is 7%, but the uncertainty range should encompass the other experimental results. We have carried out further experiments and analysis which provide substantial evidence that the 7% branching fraction from the MPIMS experiments (at 30 and 740 Torr) reflect the direct yield. In the chamber experiments, a very small signal was observed at the mass of the protonated trioxide (65), but our calculations (discussed above with reference to P7) show that it is likely that the majority of the trioxide which doesn't undergo heterogeneous loss is converted to CH_3O_2^+ and protonated methanol following protonation in the PTR-TOFMS system. Experiments have been conducted at 10, 30 and 740 Torr to investigate the formation of trioxide.

MPIMS experiments have been undertaken using three different radical precursors: $\text{XeF}_2/\text{CH}_4/\text{D}_2\text{O}$ (740 Torr) to generate CH_3OO and OD , $\text{XeF}_2/\text{CH}_4/\text{H}_2\text{O}$ (10 Torr) to generate CH_3OO and OH , and $^{13}\text{CH}_3\text{I}/\text{H}_2\text{O}_2$ (30 Torr) to generate $^{13}\text{CH}_3\text{OO}$ and OH . In all experiments, the exact mass of the side-product CH_3OOCH_3 (or the relevant isotopic analogues) originating from the self-reaction of CH_3OO , was utilized to calibrate the mass axis to determine the exact mass at which the relevant trioxide isotope should appear if formed.

In the low-pressure experiments (10 and 30 Torr), no evidence for trioxide formation is found (Supplemental Figure 13 and 14, shown below). This result is consistent with the branching fractions of 0.002 % and 0.016%, respectively, using the pressure and temperature dependent expression of Müller *et al.* At 740 Torr, where the Müller *et al.* expression predicts a yield of 9.6 % trioxide, a significant peak is detected at the exact m/z corresponding to CH_3OOD – the isotopologue of trioxide from $\text{OD} + \text{CH}_3\text{OO}$ (Supplemental Figure 15). The signal is time-resolved (Supplemental Figure 16), with an almost instantaneous production following laser photolysis – matching the initial signal of CH_3OD (from $\text{OD} + \text{CH}_3\text{OO}$), as would be anticipated from the rapid $\text{OD} + \text{CH}_3\text{OO}$ reaction. At longer kinetic times, the CH_3OD signal continues to increase (due to secondary sources from reactions such as $\text{DO}_2 + \text{CH}_3\text{O}$), whereas the trioxide signal decays, either due to removal reactions or wall loss.

We do not have a reference photoionization spectrum of the trioxide to compare with our observed signal at m/z 64. However, we have calculated the adiabatic energy (9.9 eV) at the CBS-QB3 level of theory. This calculated threshold for ionization is consistent with our observations of $\text{CH}_3\text{O}_3\text{D}$ at 11.0 eV at 740 Torr.

Our observation of the trioxide at 740 Torr in the MPIMS signal therefore confirms that trioxide is indeed formed from the $\text{OH} + \text{CH}_3\text{OO}$ reaction at higher pressures. Therefore, the trioxide will also be produced in the ambient pressure chamber experiments. As discussed earlier in this reply, we calculate that the protonated trioxide will decompose largely either to CH_3O_2^+ , or protonated methanol, depending on the site of protonation. Through our observation of trioxide in our higher pressure MPIMS experiments, and our calculations demonstrating that decomposition of the protonated trioxide will lead to artificially high methanol signal in the PTR-TOFMS experiments, we demonstrate that the observed yield of ~6-9% methanol in the MPIMS experiments is a more robust determination of the direct methanol yield from the $\text{OH} + \text{CH}_3\text{OO}$ reaction, and that the

apparent 17% yield from the PTR-TOFMS chamber experiments include contributions from the trioxide product.

Supplemental Figure 13 Integrated ion signal from the reaction of OH + CH₃OO at 10 Torr using XeF₂/CH₄/H₂O precursors measured with a photoionization energy of 11.5 eV.

Supplemental Figure 14 Integrated ion signal from the reaction of OH + $^{13}\text{CH}_3\text{OO}$ at 30 Torr using $^{13}\text{CH}_3\text{I}/\text{H}_2\text{O}_2$ precursors measured with a photoionization energy of 11.0 eV.

Supplemental Figure 15 Integrated ion signal from the reaction of OD + CH_3OO at 740 Torr using $\text{XeF}_2/\text{CH}_4/\text{D}_2\text{O}$ precursors measured with a photoionization energy of 11.0 eV.

Supplemental Figure 16 Temporal profiles of $\text{CH}_3\text{O}_3\text{D}$ (red closed circles) and CH_3OD (black open circles) from the reaction of $\text{OD} + \text{CH}_3\text{OO}$ at 740 Torr using $\text{XeF}_2/\text{CH}_4/\text{H}_2\text{O}$ precursors measured with a photoionization energy of 11.0 eV.

The phrasing in this section as to what is or is not a ‘substantial’ source of methanol is a bit misleading. Global land emissions of methanol are somewhere around 100 Tg/y and atmospheric production from peroxy radical cross and self-reactions has been estimated at ~ 40 Tg/y. So even with a methanol yield of only 7% the $\text{CH}_3\text{OO} + \text{OH}$ reaction still provides a significant augmentation (22 Tg) to the known atmospheric source of methanol, as does your upper estimate of 17% (54 Tg). It is just that both are smaller than estimated by Muller, and smaller than needed to explain atmospheric observations. **We have rephrased the sentence regarding a ‘substantial source of methanol’ to say that the methanol produced from this reaction is smaller than required to rationalize the atmospheric observations.**

Fig. 6: “black error bars represent measurement variability”. Say what that actually means. SD? **Yes, it is the standard deviation – this has been added in parenthesis.**

P11, “experimentally determined branching fraction of 6-9%”. Again, the two experimental values are 7 and 17%. You haven’t convincingly discredited the 17% value. **Please see the discussion above regarding the calculations of trioxide cations and the new MPIMS experiments, detecting the trioxide product.**

P10, “Modest changes are observed on the abundances of OH (-8%) ...”; for all the percentages listed in this section, the wording needs to specify that the percentages listed are “up to” or “maximum”, otherwise it gives a misleading impression. -8% would be quite a large global average OH change, but in fact that is the maximum value seen. **We agree this is ambiguous – we have added clarification on both page 8 and page 10 as suggested.**

P11, “using the experimentally determined methanol branching fraction of 6-9% would lead to a significant decrease in atmospheric methanol”. Earlier you said that a 7% yield increases the global methanol production rate by 3 Tg/y, so this part needs to be clearer about what you mean. Did you mean relative to the higher-yield scenarios, or specifically in remote regions? **This refers specifically to remote regions – we have added this clarification.**

Fig 6 would be easier to read if the colored lines had an intuitive color progression from low to high yield, e.g., from cool to warm colors. **We have changed the colour progression according to your suggestion.**

P15 of supplement: “a slight increment (1%) in the southern hemisphere ocean regions where NO_x is high in the model” ... this seems odd to have high NO_x in remote southern ocean areas. Please state why, is it due to fires? **The following has been added to the manuscript: The high levels of NO_x are found over the southern hemisphere continents due to biomass burning. The high levels of NO_x combined with the large emissions of VOCs (such as isoprene and monoterpenes) in the southern hemisphere continents result in the formation of organic nitrates which decompose away from source regions, releasing NO_x in the southern hemispheric oceans.**

P15 of supplement, “driven by two major NO₂ formation reactions, CH₃OO + NO (22%) and HO₂ + NO (56%)” ... It is unclear what these percentages represent. **We have rephrased to clarify that these percentages refer to the percentage of the total NO₂ formation flux.**

P15 of supplement. For the discussion of changes to the HCHO budget, it is worth also pointing out that any CH₃OH produced will also go on to produce HCHO (aside from minor deposition losses)... in other words, we don't expect very much change to the overall HCHO source regardless of the branching in these reactions. **Thank you – we have added the total amount of formaldehyde with a 17% methanol yield from the OH + CH₃OO reaction, to draw attention to this.**

Reviewer #2 (Remarks to the Author):

This is experimental study of the title reaction confirming the results of a recent theoretical study (ref 15). The experiments appear to be carefully done and well explained. The presentation seems long for a communication and I would recommend publication as a full paper. Atmospheric model incl. Figure 4-6 – could be reduced without loss of essence.

Minor

We thank Reviewer #2 for their feedback and have attempted to address all their queries and suggestions as detailed below.

Page 2, based on recent results by Brandt et al (Angew Chem 2018, 57, 3820) I would suspect the CH₃OO + RO₂ reaction to be slow compared to NO and HO₂ reactions.

Page 3, explain the phi. **We have now added the definition before the phi**

Page 6, the ~1atm, should be reworded. E.g moved till after 11% and say at 1 atm total pressure. **We have changed this as you suggest.**

Page 7/8 I find the Results () a bit odd sections. **We have replaced this with subtitles within the**

Results section.

Page 11, black triangles are hard to see. **We have made the symbols larger.**

Page 12, remove 'difficult stationary points' – what is meant by this. **We have removed this.**

Figure S3, a bit off with blue dash and black line as model, perhaps models. **Supplemental Figure 3 really just serves to demonstrate that the ratio of $[CH_2O]/[CH_3OH]$ is insensitive to the photolysis fraction of the precursors within the uncertainty parameters and is not a fit used to derive branching ratios.**

Figure S10, in all panels it seems like the model rise is not levelling off like the expt seem to do. **The reviewer has a very keen eye! At longer times, there is increasing side and secondary chemistry, which is not as well constrained. However, it is important here to note that the relevant chemistry (that of OH + CH₃OO) is over within the first 5 ms or so, and the data at the longer times commented on by the reviewer is inconsequential to determining the methanol yield.**

REVIEWERS' COMMENTS:

Reviewer #1 (Remarks to the Author):

The authors have done a commendable job in addressing the review comments. In my view the paper is much improved and the main points are now more solidly supported. It is a nice piece of work.